# Intrauterine Blood Plasma Platelet-Therapy Mitigates Persistent Breeding-Induced Endometritis, Reduces Uterine Infections, and Improves Embryo Recovery in Mares

**DOI:** 10.3390/antibiotics10050490

**Published:** 2021-04-23

**Authors:** Lorenzo G. T. M. Segabinazzi, Igor F. Canisso, Giorgia Podico, Lais L. Cunha, Guilherme Novello, Michael F. Rosser, Shavahn C. Loux, Fabio S. Lima, Marco A. Alvarenga

**Affiliations:** 1Department of Veterinary Clinical Medicine, College of Veterinary Medicine, University of Illinois Urbana Champaign, 1008 W Hazelwood Drive, Urbana, IL 61802, USA; lgseg@hotmail.com (L.G.T.M.S.); gpodico@illinois.edu (G.P.); laiscvet@gmail.com (L.L.C.); guilhermenovello@outlook.com (G.N.); falima@ucdavis.edu (F.S.L.); 2Department of Veterinary Surgery and Animal Reproduction, School of Veterinary Medicine and Animal Science, São Paulo State University (UNESP), Botucatu, Sao Paulo 18618681, Brazil; Marco.alvarenga@unesp.br; 3Ross University School of Veterinary Medicine, Basseterre PO Box 334, St. Kitts, West Indies; 4Veterinary Diagnostic Laboratory, College of Veterinary Medicine, University of Illinois Urbana Champaign, Urbana, IL 61802, USA; mrosser2@illinois.edu; 5Maxwell H. Gluck Equine Research Center, University of Kentucky, Lexington, KY 40503, USA; shavanh.loux@uky.edu; 6Department of Population Health and Reproduction, School of Veterinary Medicine, University of California, Davis, CA 95616, USA

**Keywords:** endometrium, uterine inflammation, equine, PRP, immunomodulation

## Abstract

Microorganisms, including pathogenic or opportunistic bacteria and fungi, may gain access to the uterus during breeding, and infectious endometritis plays a major role in equine subfertility. This study aimed to assess the post-breeding inflammatory response, endometrial culture, and embryo recovery of mares susceptible to persistent breeding-induced endometritis (PBIE) treated with plasma-rich (PRP) or -poor (PPP) plasma. Mares (*n* = 12) susceptible to PBIE had three cycles randomly assigned to receive intrauterine infusions of lactate ringer solution (LRS, control), or autologous PRP or PPP pre- (−48 and −24 h) and post-breeding (6 and 24 h). Mares were bred with fresh semen from one stallion. Intrauterine fluid accumulation (IUF) and endometrial neutrophils were assessed every 24 h up to 96 h post-breeding. Uterine cytokines (Ilβ, IL6, CXCL8, and IL10) were evaluated before (0 h), 6, and 24 h post-breeding, and endometrial culture three and nine days after breed. Embryo flushing was performed 8 days post-ovulation. Data were analyzed with mixed model, Tukey’s post-hoc test, and multivariate regression. PRP treatment reduced endometrial neutrophils, post-breeding IUF, and pro-inflammatory cytokines when compared to control-assigned cycles, but not significantly different than PPP. Controls had a significantly higher percentage of positive bacterial cultures (33%) in comparison to PRP-assigned cycles (0%), whereas cycles treated with PPP were not significantly different from the other groups (25%). The PRP-assigned cycles had significantly greater embryo recovery rates (83%) than the control (33%), though not significantly different than PPP (60%). Plasma infusion reduced the duration and intensity of the post-breeding inflammatory response and improved embryo recovery in mares susceptible to PBIE. Platelets incrementally downregulate PBIE and appear to have a dose-dependent antimicrobial property.

## 1. Introduction

Endometritis is the third most common disease affecting horses in the United States [1] and is the number one cause of subfertility and poor reproductive efficiency in mares [2]. Mares are classified as susceptible or as resistant to endometritis based on their ability to clear uterine infection/inflammation by 48–72 h post-breeding [3]. Persistent breeding-induced endometritis (PBIE) can be caused by infectious (i.e., bacteria and fungus) and by non-infections agents such as sperm [4]. All mares display a physiological and transient uterine inflammatory response; however, the mares deemed susceptible to PBIE have a delayed and prolonged uterine inflammatory response [5]. The persistent endometrial inflammation leads to a hostile uterine environment for the embryo entering the uterus, compromising embryo recovery and pregnancy rates [6,7]. 

Infectious endometritis plays a major role in equine subfertility; up to 25–60% of mares failing to become pregnant have bacterial uterine infection [8,9,10,11]. Microorganisms, including pathogenic or opportunistic bacteria and fungi, may gain access to the uterus during breeding. While the resistant mares respond rapidly to the presence of sperm and microorganisms, inadequate immune response may lead to persistent inflammation and infection in mares susceptible to PBIE [12]. 

Traditionally, PBIE has been treated with multi-modal therapeutics, such as a combination of uterine lavage, ecbolic agents, anti-inflammatories, and antibiotics [8]. While employing multi-modal therapeutics effectively manages PBIE, some mares, particularly aging broodmares and donor mares enrolled in embryo transfer programs for multiple years, often fail to respond to traditional therapy for PBIE (8). Furthermore, in numerous stud farms, all mares in the premises are prophylactically infused with broad-spectrum antibiotics regardless of need; such practice likely contributes to the development of antimicrobial resistance. 

In recent years, the lack of response to conventional therapy coupled with the increasing incidence of antibiotic-resistant bacteria led to the development of alternative treatments for mares suffering from PBIE [13]; these have been largely attributed in part to the indiscriminate use of antibiotics in veterinary medicine (e.g., non-select intrauterine infusion in mares). Multidrug-resistant microorganisms pose a major global threat to public health. Autologous platelet-rich plasma (PRP), a whole blood plasma with a high platelet concentration, is becoming a popular nontraditional therapy in human and veterinary medicine as an alternative to circumvent such problems. This byproduct has been used for its anti-inflammatory, regenerative, and antimicrobial properties [14,15,16,17,18]. 

Platelets, or thrombocytes, are fragments of cytoplasm derived from megakaryocytes, which are large cells present in the bone marrow [19]. The cytoplasm of platelets is subdivided into chromomere, where granules accumulate, and the hyalomere, an agranular region rich in cytoskeletal proteins [15]. Platelet granules contain numerous proteins such as fibrinogen, growth factors (e.g., transforming growth factor β, vascular endothelial growth factor), cytokines (e.g., CXCL8 and TNFα), and antimicrobial peptides (e.g., platelet factor 4, RANTES, connective tissue activating peptide 3, platelet basic protein, thymosin beta-4, fibrinopeptide A and B) [20,21]. 

In mare reproductive practice, studies showed that intrauterine infusion with PRP could improve pregnancy rates by mitigating the post-breeding inflammatory response [22,23,24,25,26], but PRP has not been critically assessed in embryo donors mares, a group of mares prone to PBIE. Platelets’ natural antimicrobial peptides are thought to be responsible for PRP’s benefit on septic arthritis in horses [27]; however, PRP’s anti-microbial beneficial properties have yet to be evaluated in mares susceptible to PBIE. 

The hypotheses of this study are that the administration of PRP to embryo donor mares susceptible to PBIE will result in fewer uterine infections and a lessened post-breeding inflammatory response in comparison with platelet-poor-plasma (PPP) or control (Lactate Ringer’s Solution LRS) treatments. Specifically, PRP therapy reduces the duration and intensity of the post-breeding inflammatory response and reduce the chances of post-breeding uterine infection leading to enhanced embryo recovery rates of mares susceptible to PBIE. This study aimed to compare PRP and PPP’s effects with control-assigned cycles on uterine microbiology, endometrial inflammation, intrauterine fluid accumulation, progesterone concentration, and endometrial receptor expression and embryo recovery rates in mares susceptible to PBIE.

## 2. Results

### 2.1. Screening Mares for Susceptibility to PBIE

After screening twenty-two mares, twelve mares were identified as susceptible and seven as resistant to PBIE [3], while three were deemed intermediate. Susceptible mares had an increased intrauterine fluid until 72 h post-sperm challenge (*p* < 0.05), while resistant mares did not (*p* > 0.05) (Appendix A). In addition, susceptible mares had greater intrauterine fluid accumulation than resistant mares at all time points post-sperm challenge (*p* < 0.05). Similarly, PMNs were more abundant post sperm-challenge (*p* < 0.05) in susceptible mares than resistant mares (Figure 1). There was an effect of time (*p* < 0.05) but no differences between mare groups for edema scores (*p* > 0.05) (Appendix A). Eight mares were classified as IIB and four mares as III in the susceptible group, whereas in the resistant mares, two were classified as I and five were classified as IIA. 

### 2.2. Platelet-Rich or -Poor Plasma

Mares susceptible to PBIE had a platelet concentration in whole blood ranging from 74.6 to 188.0 × 10^3^ platelets per µL (Table 1). There was a moderate correlation (r = 0.5) between the platelet concentration in the whole blood and the PRP’s final platelet concentration. The mean platelet concentration increased 5.2-fold in PRP compared to the whole blood count (*p* = 0.0003, Table 1). There was a reduction in RBC (266-fold) and WBC (248.5-fold) in PRP and PPP in comparison to whole blood (*p* < 0.0001). Platelet-poor plasma had a lower platelet concentration than the whole blood (3.3-fold reduction) and PRP (17.3-fold reduction) (*p* < 0.02, Table 1). Platelet viability was similar for both PRP (97.0 ± 0.7%) and PPP (97.2 ± 0.6%) (*p* = 0.79, Figure 1). The mean number of platelets infused in the uterus of mares was greater in PRP (24.9 ± 1.2 × 10^9^ platelets) than PPP (1.4 ± 0.2 × 10^9^ platelets) (*p* < 0.001). 

### 2.3. Semen Parameters

Twenty-six ejaculates were harvested and used for breeding in the study. Breeding doses and sperm parameters did not differ across cycles during the experiment (*p* > 0.05, Appendix A).

### 2.4. Intrauterine Fluid Accumulation and Endometrial Edema 

There were no differences for intrauterine fluid accumulation before treatment (−48 h) and breeding (0 h) in all cycles (*p* > 0.05, Figure 2A). Endometrial edema scores were not different among groups at 0, 24, 48, 72, and 96 h (*p* > 0.05). However, intrauterine fluid accumulation was reduced up to 96 h post-breeding in mares with assigned cycles to receive PRP when compared to mares assigned to the control cycles (*p* < 0.05). The cycles assigned to PPP were not different than PRP or control-assigned cycles (*p* > 0.05) (Figure 2A). However, at 48 h, intrauterine fluid accumulation tended (*p* = 0.09) to be reduced in the PPP assigned cycles when compared to the control cycles. At 96 h after breeding, mares in the control cycle had more intrauterine fluid accumulation than mares treated with PRP (*p* = 0.043). Two mares stopped cycling due to seasonality while assigned to PPP cycles; therefore, only ten cycles were fully completed for PPP.

### 2.5. Inflammatory Cell Counts on Endometrial Biopsy and Cytology

Endometrial cytology revealed that PRP reduced (*p* < 0.0001) the PMNs number at 24 and 72 h, as well as tended (*p* = 0.08) to reduce at 48 h after breeding when compared with similar time points in the control-assigned cycles (Figure 2B). The PPP-assigned cycles did not differ from the other cycle-assignments for the number of PMNs in cytology (*p* > 0.05). However, both PRP and PPP reduced the number of PMNs in endometrial biopsies at 6 (*p* = 0.001) and 24 h (*p* < 0.0001) post-breeding and eight days post-ovulation (*p* = 0.0047, Figure 2C and Figure 3). There were no differences in the number of lymphocytes among groups or time points (*p* > 0.05; Figure 3). Moderate correlation (r = 0.65) was observed between PMNs counted in endometrial cytology samples and endometrial biopsy samples.

### 2.6. Cytokine Concentrations in Uterine Fluid

Cytokine concentrations were assessed in a subset of mares. There were no changes in the concentration of the cytokines assessed in the present study among cycles before breeding (*p* > 0.05). Concentrations of IL1β increased 6 h post breed in all groups (*p* < 0.05, Figure 4A), and cycles treated with PRP had lower IL1β and IL6 at 24 h than control-assigned cycles (*p* < 0.05, Figure 4A, B). CXCL8 increased post-breeding in mares at the control-assigned cycle (*p* < 0.05, Figure 4C). Greater concentrations of CXCL8 were noted post-breeding (6 and 24 h) in mares assigned to control cycles compared with PRP-assigned cycle (*p* < 0.05, Figure 4C). There were no changes in IL10 concentrations in the uterine fluid across time or treatment (*p* > 0.05, Figure 4D).

### 2.7. Immunohistochemical Evaluation

Positive nuclei to PR were brown-stained, while negative nuclei were blue-stained by hematoxylin counterstain. The PR expression was intensely detected in epithelial cells and glandular epithelium; however, it was rarely detected in the stroma. There were no changes in PR expression between treatments (*p* > 0.05; Figure 5). Negative controls showed no immunoreaction.

### 2.8. Endometrial Culture and Progesterone Concentrations

Estrous cycles assigned as Control and PPP tended (*p* = 0.08) to result in a greater percentage of positive aerobic bacterial cultures at Day-2 than PRP cycles. In contrast, at Day-8, mares assigned as control had a greater percentage of positive bacterial cultures in comparison to PRP-assigned cycles (0%) (*p* = 0.0373), whereas cycles assigned to PPP were not different from the other groups (*p* > 0.05, Table 2). The concentrations of progesterone increased over time across groups (*p* < 0.0001), and PRP assigned cycles had greater progesterone concentrations at Day-8 than in the control-assigned cycles (*p* = 0.0376), but not different than PPP assigned-cycles (*p* > 0.05; Figure 6). There were no differences on progesterone concentrations between mares with positive and negative embryo flushes in the control group (*p* > 0.05) or all cycles together positive vs. negative embryo flushing (*p* > 0.05; Appendix A). 

### 2.9. Embryo Recovery Rates

Embryo recovery was greater in PRP-assigned cycles (83%) when compared with control-assigned cycles (33%) (*p* = 0.0361), whereas PPP-assigned cycles had intermediate embryo recovery rates but not different from the other assigned cycles (*p* > 0.05). Of interest, when treatment was considered as an independent variable (PRP and PPP), mares had greater fertility rates after plasma therapy (*p* = 0.0356; 72%, 16/22) than the control-assigned group. The PRP and PPP treatments increased 2.5- and 1.5-fold the embryo recovery rates compared with the control assigned cycles. In addition, when data were adjusted for the number of ovulations per cycle, PRP therapy increased embryo recovery (73.3%) compared with the control cycle (28.6%) (*p* = 0.0268). In the PPP assigned cycles, the embryo recovery rate was intermediate (53.9%), when embryo recovery rates were adjusted for the number of ovulations per cycle but not different than other cycles-groups (*p* > 0.05). 

A total of 20 successful embryo collections were attained. Four embryos were recovered from four mares in the control-assigned cycles, with two of these mares having double ovulations. Eleven embryos were recovered in PRP-assigned cycles, and seven embryos were recovered from the PPP-assigned cycles. Three mares experienced double ovulations in both plasma-assigned cycles. The overall grade quality, stage of development, and diameter of embryos collected were similar across cycle assignments (*p* > 0.05, Table 3).

## 3. Discussion

The present study was set forth to assess blood plasma therapy (rich or poor in platelets) in embryo donor mares susceptible to PBIE. Intrauterine treatment with plasma mitigated PBIE in susceptible mares as evidenced by the reduction of intraluminal and endometrial PMNs, uterine inflammatory cytokines, intrauterine fluid accumulation, and the number of positive bacterial cultures compared to control-assigned cycles. Ultimately, intrauterine PRP therapy increased the percentage of embryo recovered per flushing and per number of ovulations, likely due to its immunomodulatory and antimicrobial properties of platelets turning a hostile uterine environment into an embryo-friendly uterine environment in embryo donor mares susceptible to PBIE. 

It is unclear how PRP’s infusions improve endometrial receptivity and fertility in mares. In humans, it has been shown that PRP enhances migration and proliferation of endometrial cells [29], promotes neo-angiogenesis in the endometrium of infertile women [30], and upregulates genes involved in implantation (e.g., prostaglandin-endoperoxide synthase 2 [COX2], tumor protein p53 [TP53], estrogen receptors [ER-α and ER-ß] and progesterone receptor) [31]. It is possible that the antimicrobial and anti-inflammatory properties of PRP may act synergistically to improve the uterine environment.

Platelet-rich plasma is routinely used in equine clinical practice to treat joints, bursae, and soft tissue injuries (e.g., tendonitis, tenosynovitis, and skin wounds) [32,33,34]. Interesting recent findings demonstrated potent in vivo antimicrobial properties against bacterial growth in equine synovial fluid [17] and in infected skin wounds of dogs [16]. In mare reproductive practice, autologous PRP administration before breeding to barren mares susceptible to endometritis improved pregnancy rates [24,25]. In addition, PRP was shown to downregulate endometrial transcripts for interleukins IL1β, IL6, and CXCL8 and consequently mitigated post-breeding inflammatory response [23,25]. In addition, other therapies used to mitigate PBIE (e.g., administration of dexamethasone or *Mycobacterium phlei* cell wall extract) also reduce the endometrial expression of pro-inflammatory cytokines (e.g., IL1 and IL6) [35,36]. Our findings herein with embryo donor mares corroborate with these previous studies with broodmares, highlighting PRP’s benefits for uterine immunity and reproductive performance. However, the mechanisms of action of PRP in the uterus remain to be fully elucidated. It is not well known if this downregulation in inflammatory markers post-breeding in mares treated with PRP is due to the anti-inflammatory or the antimicrobial properties of PRP, since bacterial uterine infection upregulates inflammatory cytokines in mares [36,37,38,39]. 

Studies in other body systems (i.e., human chondrocytes) suggested that PRP acts by inhibiting the translocation of nuclear factor-kappa B (NF-kB) to the nucleus [40,41], as hepatocyte growth factor contained in PRP prevents the migration of NF-kB from the cytosol to the nucleus [40]. The downstream effects of NF-kB include activation of pro-inflammatory cytokines, chemokines, and COX-2 that regulate the inflammatory signals [42]. Cytokines and COX-2 act as mediators between cells modulating the acute inflammatory response [43]. Notably, PRP was shown to suppress COX-2 in the endometrium of mares susceptible to PBIE [25]. 

Mares susceptible to PBIE have pronounced expression of pro-inflammatory and reduced anti-inflammatory cytokines to regulate acute inflammation when compared to resistant mares [35,44]. IL10 is an anti-inflammatory cytokine able to reduce the transcription of pro-inflammatory cytokines by inflammatory cells [45]. Despite the lack of change in luminal IL10 in the present study, intrauterine infusion of PRP downregulated the pro-inflammatory cytokines (e.g., IL1β, IL6, and CXCL8), which might prevent uncontrolled inflammatory response. Specifically, the endometrial mRNA abundance of IL1β, IL6, CXCL8, and TNFα are higher in susceptible than resistant mares, even before contact with the antigen [35], therefore justifying the strategy used herein to start treating mares before breeding, but it remains to be determined if treating mares with PRP both before and post-breeding have additional immunomodulatory benefits than treating mares only pre- or post-breeding. Infusion of PRP pre- or post-breeding was reported to result in similar downregulation of the post-breeding inflammatory response in mares [25]. 

The chemoattraction of PMNs is mainly mediated by CXCL8 [44,46]. One study demonstrated a reduction in transcripts for endometrial CXCL8 in PRP-treated mares [23]. In the present study, a reduction in luminal CXCL8 concentration at 6 and 24 h post-breeding was also observed in mares treated with PRP; this can explain the suppression of inflammatory cells observed in the present study. Future studies should be carried out to determine the mechanistic interactions between the endometrium of mares susceptible to PBIE and plasma and platelets. 

Intrauterine infusion of bacteria upregulates inflammatory cytokines in mares [36,37,38,39]. Pathogenic microorganisms may gain access to the uterus during breeding, and the contamination of the uterus with bacteria increases the inflammatory reaction and delays the healing process by causing ongoing inflammation [8,36,38]. Of interest, none of the PRP-assigned cycles had a positive bacterial culture at two- and eight-days post-ovulation, whereas 25% and 42% control-assigned cycles had positive bacterial cultures two days post-ovulation and on the day of embryo flushing, respectively. On the other hand, the PPP-assigned cycles, which had 17 times fewer platelets than PRP, had three (30%) estrous cycles with a positive aerobic culture two- and eight-days post-ovulation. Platelets contain Growth factors and antimicrobial peptides [47,48], which may be responsible for the findings of the present study. Peptides in platelets may explain the dose-dependent antimicrobial activity obtained here and in another study using horse’ synovial fluids as a model [17]. The peptides contained in PRP have antimicrobial activity against *Staphylococcus aureus*, *Escherichia coli*, and *Klebsiella pneumoniae* [18,49,50], which are known causes of endometritis in mares [8]. While platelet-derived antimicrobial peptides have not been described in horses, it is reasonable to suggest that horses are like to other species. Interestingly, an early study indicated that the addition of autologous plasma to antibiotic therapy after breeding lactating and barren mares could improve pregnancy rates per cycle compared with antibiotics alone [51]. Although the apparently greater concentration of platelets in PRP is correlated with increased antimicrobial potential, the plasma components may also play an important role in antimicrobial activity of PRP [47]. 

Plasma contains the complement system, an element of the immune system essential for humoral defense mechanisms against infectious agents. Bacterial cell lysis and leukocyte recruitment are associated with the activation of the complement cascade, and apparently, the plasma components (complement factors) play an important role for the antimicrobial activity of platelet concentrates [47,52]. In the present study, we demonstrated that both PRP and PPP alleviated the post-breeding endometrial inflammatory response in mares. Our results are consistent with an in-vitro study with human endometrial cells in which PRP and PPP had similar immunomodulatory capabilities [29]. The lack of differences between PRP and PPP for some of the endpoints assessed in the present study could simply mean the number of platelets contained in PPP herein was enough to elicit equivalent immunological properties to PRP. The minimal number of platelets per infusion or the number of infusions has not been determined yet. It is possible that after four infusions with PPP, a minimum number of platelets was achieved in the uteri of mares susceptible to PBIE in the present study. 

Bacterial uterine infection induces endometrial inflammation [8,36]. The inflammatory cascade during the peri-ovulatory period and early diestrus may induce excessive production of PGF2α, which can negatively affect the luteal function and progesterone concentrations, and its reduction is thought to affect pregnancy rates in mares [53]. Plasma progesterone concentrations at Day-8 post-ovulation of mares in the control cycle were lower than those treated with PRP or PPP. It is unknown if progesterone concentrations were reduced in the control-assigned cycles due to chronic inflammation or due to the fact that pregnancy rates were greater in the plasma assigned cycles. Notably, there were no differences in progesterone concentrations between cycles with a positive and negative embryo flushes cycles in the control group or all cycles together positive vs. negative embryo flushes. A previous study showed that mares becoming pregnant had greater progesterone concentrations five days post-ovulation than mares bred but not becoming pregnant [54]. In addition, endometrial infection with pathogenic bacteria (e.g., *Streptococcus zooepidemicus*) has been reported to decrease progesterone concentrations in the early diestrus of mares [55]. In addition, intrauterine PRP therapy upregulated progesterone receptors in bovine endometrial cells [31], which led the authors to suggest that intrauterine therapy with PRP may improve cow’s fertility [31,56]. However, there were no changes in PR expression in the endometrium of mares treated with PRP or PPP in the present study. This could simply mean a difference between species or the small number of mares used herein, or due to the fact that the embryos were collected eight days post ovulation, it is possible that differences in progesterone receptor could have been seen if mares were sampled later in diestrus/pregnancy. Thus, it remains to be determined if PRP therapy modulates the progesterone receptor. 

The assessment of post-breeding intrauterine fluid accumulation is an essential clinical parameter associated with endometritis in mares [8]. After immune response activation, the pro-inflammatory cytokines, including chemokines (i.e., TNF, IL1, IL6, and CXCL8), are released, and vascular endothelial cell activation occurs. Constriction of arterioles and dilation of venules results from acute inflammation, which increases vascular permeability and exudates leakage to the interstitium, causing edema and intrauterine fluid accumulation [57]. Pre-breeding intrauterine fluid accumulation not associated with breeding is indicative of susceptibility to PBIE [58]. Four mares in the present study exhibited intrauterine fluid accumulation before insemination; however, all of them had negative aerobic culture and negative endometrial cytology before each treatment. Intrauterine fluid accumulation in these cases originates from endometrial gland secretion and transudation [59], and it is usually associated with the estrus phase with increased endometrial secretion and edema [60]. In the present study as well as previous studies, PRP therapy also showed a reduction in the intrauterine fluid accumulation in barren mares and mares with chronic degenerative endometritis [22,24], which suggests a beneficial effect of this therapy for mares with delayed uterine clearance. In the present study, all mares had uterine lavage performed at various time points for sampling. This procedure alone likely mitigated the inflammatory response in mares. However, since mares were equally submitted to this therapy across cycle groups, the effects of uterine lavage equally affected each cycle. It is possible that if no uterine lavage were performed, the results could have been even more contrasting between cycle groups. Uterine biopsy was performed across groups for sampling purposes. It is possible that endometrial biopsy caused further inflammation; however, since all groups were equally sampled, the effects were properly distributed across groups.

## 4. Materials and Methods

All experimental protocols conducted in the present study were approved by the Institutional Animal Care and Use Committee of the University of Illinois Urbana-Champaign under protocol # 19141. The study was carried out from May to December 2019 at the University of Illinois Veterinary Teaching Hospital.

### 4.1. Screening Mares for Susceptibility to PBIE 

Twenty-two light-breed mares (11.7 ± 1.2, range 5 to 18 years-old) belonging to the University of Illinois were screened for susceptibility to PBIE as previously described [3]. All the mares were previously used as embryo donors in the principal investigator’s research program for at least two consecutive breeding seasons. 

Eight ejaculates were collected from two Quarter Horse stallions (*n* = 4/each) with a Missouri artificial vagina (Nasco, Fort Atkinson, WI, USA). After collection, raw semen was centrifuged at 600× *g* for 10 min, and the supernatant (seminal plasma) was discarded. The sperm pellet was pooled, concentration was assessed, packed (Whirl-Pak, Nasco) at 2 billion sperm in 20 mL of PBS, and stored at −20 °C. Two freezing–thawing cycles were performed to obtain killed sperm before each use in the screening test. 

Before the screening, each mare should have no signs of pre-existent uterine infections or inflammation as determined by endometrial aerobic bacterial culture, endometrial cytology, and no intra-uterine fluid accumulation detected on transrectal ultrasonographic examination. Mares had transrectal palpation and ultrasonography examination performed every other day until a 30 mm follicle was detected. Thereafter, mares were examined daily until the follicle reached ≥35 mm in the presence of endometrial edema score ≥ 1. Endometrial edema was scored on each transrectal ultrasonographic examination on a scale of 0 (absent) to 4 (max). Ovulation was induced with 500 µg of histrelin acetate (Botupharma Sao Paulo, Brazil), and mares had 2 billion killed sperm infused into the uterus. Endometrial cytology and ultrasound examination of the reproductive tract was performed daily up to 96 h. Endometrial biopsy and aerobic bacterial culture samples were obtained immediately before and 96 h after the sperm challenge. Thereafter, mares were classified as susceptible to PBIE if they had intrauterine fluid accumulation (column ≥ 2 cm), a positive endometrial cytology (≥3–5 PMNs/hpf), and/or a positive aerobic culture at 96 h after sperm challenge. In addition, mares classified as susceptible to PBIE had IIB or III scores in the Kenney and Doig classification [61]. 

Mares that failed to display intrauterine fluid accumulation, had negative endometrial cytology at 48 h, and negative aerobic bacterial culture at 96 h after the sperm challenge were deemed resistant to PBIE. To be classified as resistant to PBIE, mares also had endometrium classified as I or IIA [61]. A mare fitting one criterion of susceptibility to PBIE but not another was deemed as intermediate. 

### 4.2. Experimental Design

Mares screened in the preliminary study classified as susceptible to PBIE (*n* = 12, 14.2 ± 0.8, range 8 to 18 years old) were enrolled. All mares were randomly assigned into three groups in a crossover design: PRP, PPP, and LRS. After the mare had an estrous cycle assigned in one of the groups, the mare had a washout cycle to minimize the previous estrous cycle’s potential carryover effects. Before each treatment cycle, all mares needed to have a negative aerobic bacterial endometrial culture, along with an endometrial cytology free from inflammation during estrus. Otherwise, mares with positive endometrial cytology or aerobic bacterial culture were treated as needed and had an additional washout cycle. During each washout cycle, mares needed to fit similar criteria before being assigned in the next treatment cycle.

For all three groups, each intrauterine infusion (four in each estrous cycle) consisted of 40 mL of LRS, PRP, or PPP. Transrectal palpation and ultrasonography examination were performed three times a week, and prostaglandin F2alpha (dinoprost 5 mg/animal i.m., Lutalyse^®^, Zoetis, Parsippany, NJ, USA) was administered if a CL was present to bring mares back into estrus. 

Once a preovulatory follicle was detected (≥33 mm in the presence of endometrial edema score ≥ 1), endometrial culture, cytology, low-volume uterine lavage, and biopsy were obtained. In addition, a plasma sample was obtained via venipuncture for assessment of progesterone concentrations with an immunoassay. After sampling, mares were submitted to uterine lavage with 2 L of LRS. Immediately after uterine lavage, mares were enrolled in one of the three groups to receive an intrauterine infusion of 40 mL of one of the treatments described above. Before each intrauterine procedure, the perineum was aseptically prepared using iodine scrub, rinsed with clean water, and dried with a paper towel. In the subsequent day, each mare received histrelin acetate (500 µg, i.m.) for induction of ovulation followed by intrauterine treatment. On the next day, mares were inseminated with ~2 billion of fresh semen. Six hours post-breeding, low-volume uterine lavage and an endometrial biopsy were obtained, and the uterus was flushed with 2 L of LRS and infused with 40 mL of one of the treatments. The next day, transrectal palpation and ultrasonography examination, endometrial cytology, low-volume uterine lavage, and biopsy were performed. Thereafter, the uterus was flushed with 2 L of LRS and infused with the respective cycle-assignments. 

Transrectal palpation and ultrasonography examination was performed daily to confirm ovulation and to assess the endometrial edema and intrauterine fluid accumulation until three days post-ovulation. If intrauterine fluid accumulation was present, the height and the width of the fluid column (mm2) were measured with the ultrasound caliper function at the uterine bifurcation. Oxytocin (20 units, i.m.) was administered twice daily after each intrauterine infusion (6 and 12 h post-treatment) and then morning and afternoon 12 h apart from 48 to 96 h post-breeding. Endometrial cytology was assessed daily until two days after ovulation was confirmed. Mares failing to ovulate by 24 h post-insemination had the estrous cycle discarded, submitted to a washout cycle, and re-assigned back in the same treatment group.

Embryo flushing was performed eight days after ovulation (Day 8), with 4 L of LRS. All embryos recovered were measured and graded for development (e.g., blastocyst or expanded blastocyst) and quality [28]. In grade 1, the embryo had a spherical shape, uniform size of blastomeres, color, and texture, with no visible abnormalities. A grade 2 embryo could have slight irregularities in shape, size of blastomeres, color or texture, and could present some extruded blastomeres. Grade 3 embryo could have a large percentage of extruded blastomeres, partial collapse of blastocele, or moderate shrinkage of trophoblast from zona pellucida. Grade 4 embryo were those with varying advanced stages of degeneration and irregularities. Immediately after embryo flushing, each mare had an endometrial biopsy collected. Thereafter, prostaglandin F2alpha (5 mg, dinoprost, i.m.) was administered to each mare to return to estrus. 

### 4.3. Preparation of PRP and PPP

Immediately before intrauterine infusion of PRP or PPP, 450 mL of blood was collected from each animal through a venipuncture of the jugular vein using an 18G needle, into a blood transfusion bag (Jorgensen Labs, Loveland, CO, USA) containing 63 mL of citrate-phosphate-dextrose solution with adenine as an anticoagulant. Four hundred milliliters of whole blood were split into eight 50-mL tubes, and the samples were centrifuged at 400× *g* for 15 min. The supernatant was transferred into 15-mL conical tubes and again centrifuged at 1000× *g* for 10 min. After the second centrifugation, 2.5 mL of plasma at the bottom of each tube was preserved and used as PRP, while the supernatant was used as PPP. The concentration of platelet and white and red blood cells were determined in whole blood, PRP, and PPP samples using manual counting with a hemocytometer at the University of Illinois Veterinary Diagnostic Laboratory. 

### 4.4. Assessment of Platelet Viability

Platelet viability was assessed using a full-spectrum detector based (filter-less) Cytek Aurora Flow Cytometer (Cytek Biosciences Inc., Fremont, CA, USA) adapted from protocols published elsewhere [62,63]. Immunolabeling of CD41/61, a platelet-specific antigen, was carried out with a primary (CD41/61, monoclonal antibody CO.35E4, #MA5-28370 Invitrogen, Life Technologies Corporation, Carlsbad, CA, USA) and a secondary antibody conjugated with a fluorochrome (goat polyclonal anti-mouse IgG conjugated with R-phycoerythrin; #P-852, Invitrogen, Life Technologies Corporation, OR, USA). In addition, the association with zombie green (#423112 Biolegend, San Diego, CA, USA), a dye that binds to cytoplasmic amines, was used to assess the plasma membrane integrity of platelets. The working solution of Zombie Green was prepared with the dilution in PBS at a 1:100 ratio. Briefly, an aliquot of PRP or PPP was diluted 1:20 in Tyrode’s media (134 mM NaCl, 12 mM NaHCO_3_, 2.9 mM KCl, 0.34 mM Na_2_HPO_4_, 1 mM MgCl_2_, 10 mM HEPES; pH 7.4) deprived from calcium chloride. Thereafter, the sample was incubated with Zombie Green working solution (1:1) and anti-CD41/61 antibody (1:200) for 30 min at room temperature in the dark. 

Samples were washed (1500× *g*, 15 min), resuspended in Tyrode’s media, and further incubated with goat anti-mouse IgG (1:100) for 30 min at room temperature in the dark. Samples were washed (1500× *g*, 15 min), resuspended in 200 µL of the same Tyrode’s media, and immediately analyzed. This panel identified four different populations (1) intact platelets (CD41/61+ with low zombie green intensity), (2) damaged platelets (CD41/61+ with high zombie green intensity), and (3) debris (CD41/61 negative with high or low zombie green intensity). The analysis was concluded when at least 1,000,000 fluorescent gated events or 150 µL of the sample were assessed. Single-stain controls and compensation were used to unmix the signals. Heat-treated platelets (75 °C, 15 min) served as a positive control for damaged platelets. Zombie Green and R-phycoerythrin were excited and detected with a 488 nm and 575 nm fluorescence detector, respectively. Data were analyzed with the software FlowJo (V10.6.2, BD Life Sciences, Franklin Lakes, NJ, USA); the percentage of events in each population was calculated, and manual compensation was applied as needed.

### 4.5. Semen Collection and Processing 

Semen from one fertile Quarter Horse stallion housed at the University of Illinois Urbana-Champaign, IL, USA, was used for all inseminations. Semen was harvested with a Missouri artificial vagina, with the gel-free semen fraction assessed for sperm concentration in a Nucleocounter SP-100 (Nucleocounter SP-100, Chemometek, Lillerød, Denmark) following the manufacturer instructions. Briefly, 50 µL of semen was diluted in 5 mL of lysis buffer (Reagent S100, ChemoMetec, Denmark) and loaded into the cassettes before the assessment. 

The sperm motility parameters were assessed using computer-assisted sperm analysis (CASA) using default settings recommended by the manufacturer (Spermvision, Minitube of America, Verona, WI, USA) for equine sperm. The CASA’s preset values were static cell area 14–80 µm^2^; straightness threshold for progressive motility 90%; average path velocity threshold for static cell < 9.5 µm/s; cell intensity 10^6^; and light-emitting diode illumination intensity 1800–2550. Each sample was incubated for 10 min at 37 °C before each evaluation. A small aliquot (10 µL) of extended semen was placed on a pre-heated slide with a coverslip for the assessments. The percentage of total percent and progressive sperm motility was recorded for descriptive purposes. Each insemination consisted of ~2 billion sperm. 

### 4.6. Aerobic Bacterial Culture 

For aerobic endometrial cultures, the samples were obtained using double-guarded swabs (Jorgensen Labs, Loveland, CO, USA) vaginally inserted into the uterus covered with a sterile sleeve at pre- (48 h) and post-breeding (72 h). In addition, a sterile cotton-tip swab was inserted in the recovered fluid obtained from the cup filter immediately after the embryo flushing and submitted to aerobic culture. After collection, swabs were plated in the chromogenic agar (Spectrum CS Culture System, Vetlab Supply, Inc., Palmetto Bay, FL, USA). Plates were incubated at 37 °C for 48 h. Bacterial growth was identified with a matrix-assisted laser desorption ionization time-of-flight mass spectrophotometer at the University of Illinois Veterinary Diagnostic Laboratory. 

### 4.7. Endometrial Cytology

Endometrial exfoliative cytology was performed pre- (48 h) and post-breeding (24, 48, and 72 h). The samples were obtained using a disposable cytobrush (Jorgensen Labs, Loveland, CO, USA). Briefly, the cytobrush was inserted manually through the vagina and cervix into the uterus using a sterile sleeve. After collection, the slides were prepared and stained by Diff-Quick (Siemens Healthcare Diagnostics Inc., Deerfield, IL, USA). The samples were then microscopically examined under 400× objective, and the number of inflammatory cells in ten higher power fields were counted. An evaluator blinded of the treatment assessed all the samples. Positive endometrial cytology was defined as ≥3–5 PMNs per high-power-field.

### 4.8. Low-Volume Lavage and Multiplex Immunoassay

Low-volume lavage involves infusion of 120 mL of PBS into the uterus, and then the fluid was retrieved by gentle rectal manipulation of the uterus. The recovered uterine fluid was placed in two 50 mL tubes and centrifuged at 400× *g* for 20 min at 5 °C, with the supernatant recovered and centrifuged again at 2000× *g* for 20 min at 5 °C to remove cellular debris. After the second centrifugation, the supernatant was saved and stored at −80 °C for proteomic analyses.

Uterine fluid interleukins from a subset of mares (*n* = 6) randomly selected were analyzed using an equine-specific multiple sandwich immunoassay based on flowmetric MILLIPLEX MAP^®^ technology in accord with the workflow previously published [64]. The detection level was defined as the signal-to-noise-ratio (limit of detection) divided by the square root of 2 (1.6 pg/mL). Data points below the limit of detection were not considered. Concentration of IL1α, IL1β, IL6, CXCL8, IL17α, IL10, MCP-1, and RANTES in uterine fluid obtained by low-volume uterine lavage were measured. Of these, IL-17a, MCP-1, and RANTES were below the limit of detection for all samples and thus were not considered further. IL-1α was detected in only two samples with levels only minimally over the limit of detection and was therefore excluded as well. 

Each sample of uterine fluid was measured undiluted, and calibration curves for these plates were prepared in assay buffer, as previously described by Skogstrand et al. [65]. Additionally, samples of interleukins were measured undiluted and standards were prepared with the serum matrix added to all standards and quality controls, following the guidelines of the manufacturer and as previously described by Fedorka et al. (2019) [66]. The mean of intra- assay coefficients of variation weas 3.2%. 

### 4.9. Endometrial Biopsy: Inflammatory Cell Count and Immunolabelling for Progesterone Receptors

Biopsy samples were taken at the cranial uterine body using a sterilized alligator jaw biopsy forceps. After collection, biopsies were immediately fixed in 10% neutral buffered formalin and embedded in paraffin for histological evaluation. Tissue was sectioned at 5 µm thickness and stained with hematoxylin and eosin. For each sample, an evaluator blinded of treatment counted the number of PMNs and lymphocytes from five randomly selected high-power fields at 400× magnification. The averages were recorded and compared across groups.

A subset of mares (*n* = 4) was used for immunohistochemistry analyses for progesterone receptors pre-breeding (−48 h) and eight days post-ovulation (D8, day of embryo flushing) following manufacturer’s recommendations (Vectastain Standard Elite; Vector Laboratories, Inc., Burlingame, CA, USA). For immunohistochemistry, tissue sections (5 µm) were prepared on poly-l-lysine-coated glass slides to detect the presence of progesterone receptors (PR). The immunohistochemical staining of all samples was performed using the avidin–biotin–peroxidase complex procedure with a commercial immunoperoxidase kit (Vectastain Standard Elite; Vector Laboratories, Inc., Burlingame, CA, USA). Tissue sections were immersed in a pre-heated solution at 94 °C of Dewax and HIER Buffer H (Thermo Fischer Scientific, Lab Vision Corporation, Fremont, CA, USA) diluted 1:15 with deionized water for 40 min. This solution is designed to simultaneously dewax and perform heat-induced epitope retrieval. Endogenous peroxidase was blocked using 1% hydrogen peroxide in Tris buffer for 45 min. Sections were incubated for 18 h at 4 °C with anti-PR (#MA1-12626, Thermo Fischer Scientific, Lab Vision Corporation, Fremont, CA, USA) mouse monoclonal antibodies diluted 1:400. After incubation with the secondary biotinylated anti-mouse immunoglobulin (diluted 1:200; Vector Laboratories, Inc.) for 30 min, the avidin–biotin–peroxidase complex method (Vector Laboratories, Inc.) was performed. Positive staining was visualized with 3.3-diaminobenzidine-4 HCl (Vectastain, Vector Laboratories, SK-4100), and nuclei were counterstained with Mayer’s hematoxylin. Diluent negative control sections were produced by the omission of the primary antibody.

Slides were evaluated by a blinded operator at 200× magnification using a microscope (OLYMPUS BX51, Olympus, Tokyo, Japan) coupled with a camera and the software ProgRes C14 PLUS (Jenoptik, Jena, Germany). Five randomly selected areas were evaluated in each section. The percent and the intensity of PR immunoreactivity (brown stained) cells were assessed separately in the surface epithelial cells, glandular epithelium, and stroma. The intensity was scored as 1, weak positive staining; 2, moderate positive staining; and 3, intense positive staining.

### 4.10. Progesterone Assay

Plasma samples were collected from the jugular vein once a preovulatory follicle was detected (D3; ≥33 mm in the presence of endometrial edema), and 48 h (D2) and eight days after ovulation (D8) for determination of plasma progesterone concentrations. The concentrations of progesterone were assessed with a chemiluminescence platform (Immulite 1000 Siemens Medical Solution USA, Inc.). The intra-assay coefficient of variation was 2.6%, and sensitivity was 0.1 ng/mL.

### 4.11. Statistical Analyses 

Data analyses were carried out with GraphPad Prism 8.0.1. (GraphPad Software, San Diego, CA, USA). Luminex data were log (log10) transformed for normalization. Data were evaluated with a mixed model and Tukey’s post hoc test. The percentage of embryo flushes with at least one embryo, the number of embryos per ovulations, and the number of positive bacterial cultures were assessed using multivariate regression analysis. Significance was set at *p* ≤ 0.05 for all tests, and a statistically significant tendency was determined with 0.05 < *p* < 0.1. All data are presented as mean ± SEM. The degree of linear correlation between PMNs counting in endometrial cytology and biopsy, as well as platelet concentration in the whole blood and the PRP, was tested using Pearson correlation. A strong coefficient of correlation was defined as r > 0.7, moderate as 0.5 ≤ r ≤ 0.7, and a weak correlation when r < 0.5.

## 5. Conclusions

In conclusion, the current study suggests that PRP may have antimicrobial properties, and intrauterine infusion of PRP reduce the chances of uterine infection after breeding in mares as indicated by the absence of positive bacterial culture in PRP-assigned cycles. In addition, the intrauterine treatment with blood plasma therapy (rich or poor in platelets) mitigates the post-breeding uterine inflammatory response of embryo donor mares susceptible to PBIE. The apparent improved immune response is likely one of the major factors contributing to enhancing embryonic survival and consequent greater embryonic recovery obtained herein. Administration of blood plasma as prepared herein improved plasma progesterone concentrations in mares susceptible to PBIE. Finally, as described herein, plasma infusions can be used as an alternative method to manage embryo donor mares susceptible to PBIE, and PRP may augment antibiotic therapy in broodmare’s practice. 

## Figures and Tables

**Figure 1 antibiotics-10-00490-f001:**
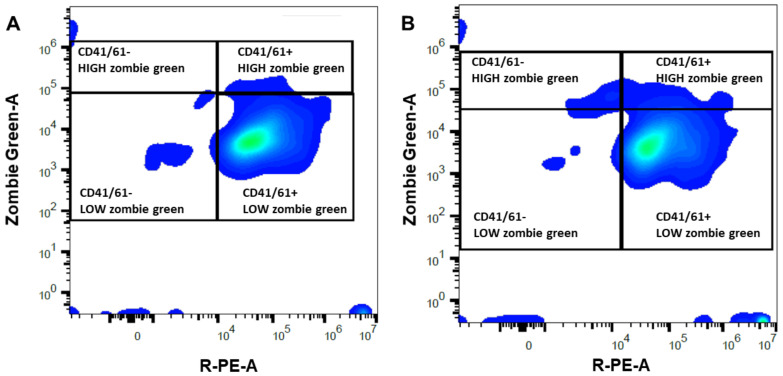
Representative density plot for flow cytometric analyses performed on platelet-rich (**A**) and -poor (**B**) plasma obtained from a mare susceptible to persistent-breeding induced endometritis. Platelets were identified with a primary (mouse monoclonal antibody anti CD41/61) and secondary antibody anti-mouse IgG conjugated with a fluorochrome (PE, R-phycoerythrin, *X*-axis) and their membrane integrity was assessed with Zombie Green (*Y*-axis). The right quadrants enclosed CD41/61 positive events, presumptively corresponding to platelets, with high (damaged membrane) or low (intact membrane) Zombie Green signal. The left quadrants included CD41/61 negative events, likely debris, with high or low Zombie Green signal.

**Figure 2 antibiotics-10-00490-f002:**
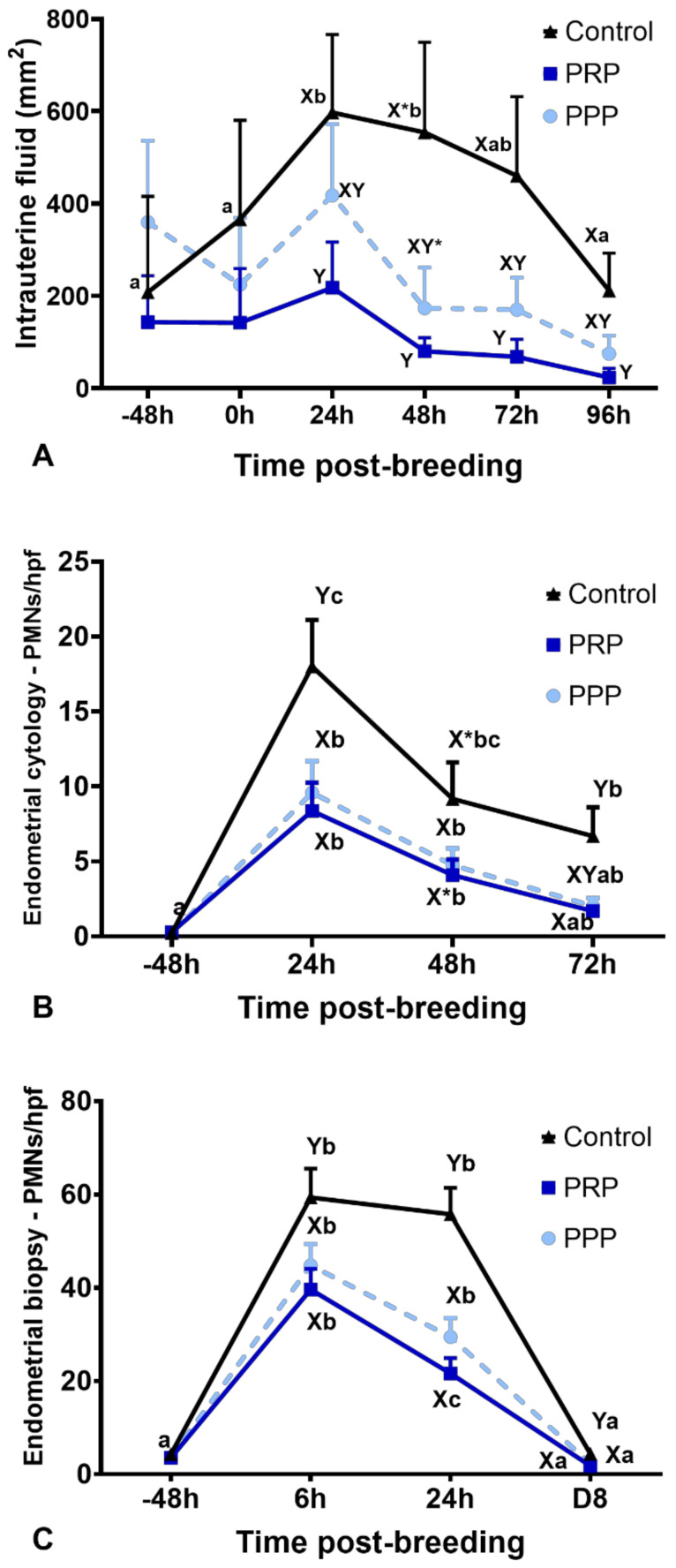
Post-breeding inflammatory response in mares susceptible to persistent breeding induced endometritis. Assessments were carried out from pre- (early onset of estrus) and post-breeding, and on the day of embryo flushing (eight days post-ovulation, D8): (**A**) Intrauterine fluid accumulation pre- and post-breeding; (**B**) mean polymorphonuclear cells (PMNs) assessed in five high-power fields (hpf) in endometrial cytology, and (**C**) mean PMNs counts assessed in five hpf in endometrial biopsies. Mares (*n* = 12) had estrous cycles (*n* = 34) assigned to receive four intrauterine infusions with Lactate Ringer’s Solution (Control, *n* = 12), platelet-rich (PRP *n* = 12), or -poor plasma (PPP *n* = 10) in a crossover design. Different superscripts denote the effects of time (^a,b,c^) and differences between groups (^X,Y^) (*p* < 0.05). Asterisk (*) denotes tendency (0.05 < *p* < 0.1).

**Figure 3 antibiotics-10-00490-f003:**
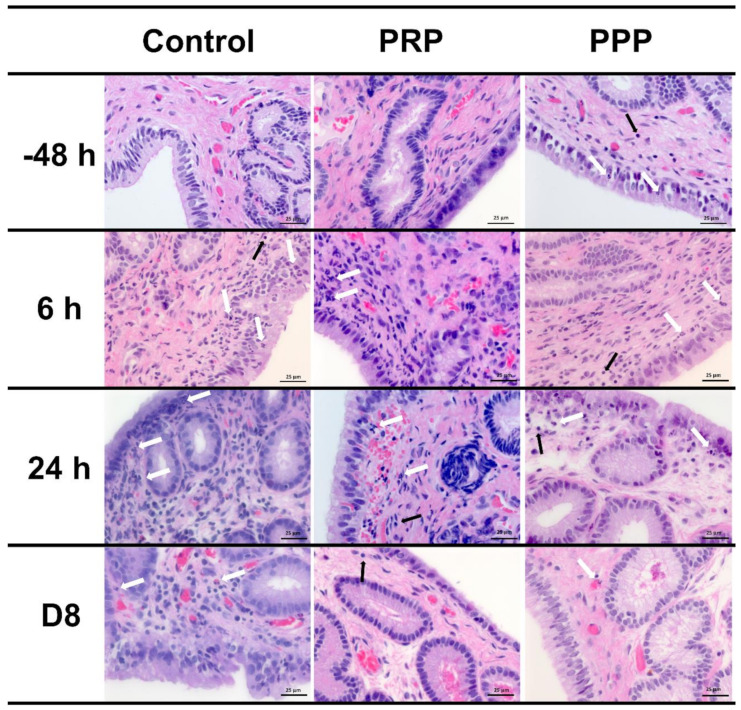
Representative images of H&E-stained endometrial biopsies collected from mares susceptible to persistent breeding-induced endometritis pre- (−48 h) and post-breeding (6 and 24 h), and on the day of embryo flushing (eight days post-ovulation, D8). Mares were treated with four intrauterine infusions of Lactated-Ringer’s solution (Control), platelet-rich (PRP), or -poor plasma (PPP). White arrows indicate neutrophils and black arrows indicate lymphocytes. Magnification ×400.

**Figure 4 antibiotics-10-00490-f004:**
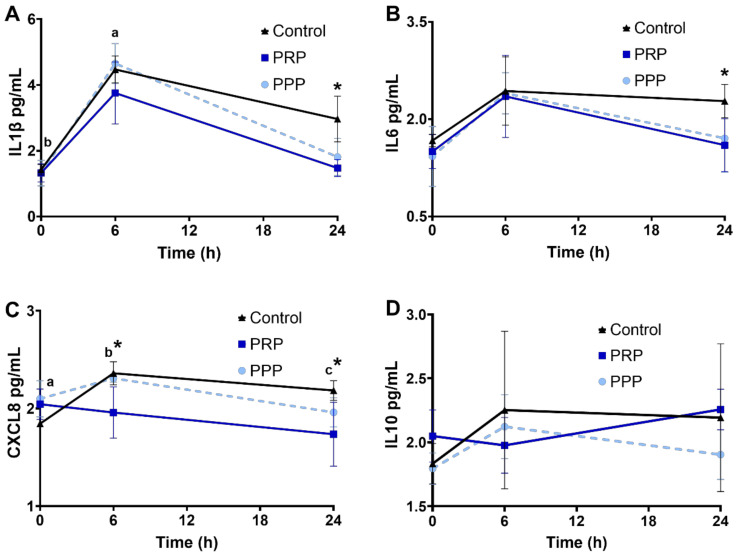
Concentrations (log10) of cytokines ((**A**), IL1β; (**B**), IL6; (**C**), CXCL8; (**D**), IL10), in uterine fluid of mares susceptible to persistent breeding induced endometritis. Assessments were carried out from pre- (0 h, early onset of estrus), 6, and 24 h post-breeding. Mares (*n* = 6) had estrous cycles (*n* = 18) assigned to receive four intrauterine infusions with Lactate Ringer’s Solution (Control, *n* = 6), platelet-rich (PRP *n* = 6), or -poor plasma (PPP *n* = 6) in a crossover design. Different superscripts denote the effects of time (^a,b,c^). Asterisk (*) denotes difference between control- and PRP-assigned cycles (*p* < 0.05).

**Figure 5 antibiotics-10-00490-f005:**
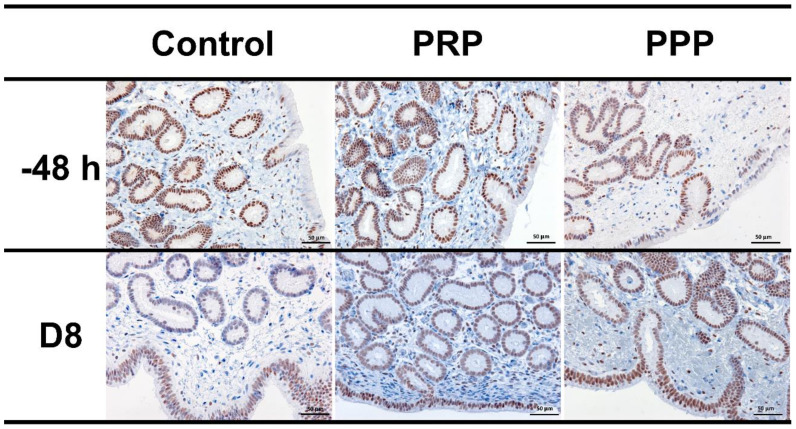
Representative images of immunohistochemical PR expression in uterine biopsies (200× magnification) from mares susceptible to persistent breeding-induced endometritis pre-breeding (−48 h) and on the day of embryo flushing (eight days post-ovulation, D8). Mares were treated with four intrauterine infusions of Lactated-Ringer’s solution (Control), platelet-rich (PRP), or -poor plasma (PPP). Positive nuclei to PR were brown-stained, while negative nuclei were blue stained by hematoxylin counterstain.

**Figure 6 antibiotics-10-00490-f006:**
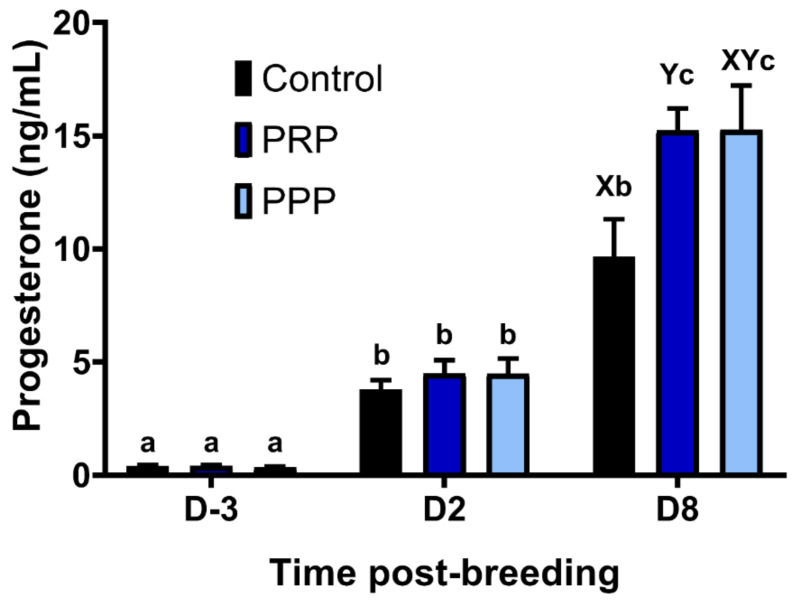
Circulating progesterone concentrations pre- and post-ovulation in mares susceptible to persistent breeding induced endometritis. Mares (*n* = 12) had estrous cycles (*n* = 34) assigned to receive intrauterine infusions with Lactate Ringer’s Solution (LRS) (Control, *n* = 12), platelet-rich (PRP *n* = 12), or -poor plasma (PPP *n* = 10) in a crossover design. Assessments were carried out from the early onset of estrus (D-3), 48 h post-ovulation (D2) and on the day of embryo flushing, eight days post-ovulation (D8): Different superscripts denote the effects of time (^a,b,c^) and differences between groups (^X,Y^) (*p* < 0.05).

**Table 1 antibiotics-10-00490-t001:** Platelets, red blood cells (RBC) and white blood cells (WBC) in the whole blood (WB), platelet-rich (PRP), and platelet-poor (PPP) plasma obtained from 96 venipunctures from mares susceptible to persistent breeding-induced endometritis. Each mare cycle was assigned in a crossover-designed to either PRP (*n* = 12), PPP (*n* = 10), or control (*n* = 12) group.

	WB	PRP	PPP
Platelets (10^3^/µL)	119.9 ± 30 ^b^	622.9 ± 144 ^a^	36.0 ± 25 ^c^
RBC (10^6^/µL)	6.4 ± 0.9 ^a^	0.02 ± 0.022 ^b^	0.01 ± 0.011 ^b^
WBC (10^3^/µL)	5.6 ± 1.2 ^a^	<0.001 ^b^	<0.001 ^b^

Mean ± SEM. Different superscripts (a,b,c) denote differences among columns within rows (*p* < 0.05).

**Table 2 antibiotics-10-00490-t002:** Aerobic culture results of aerobic obtained Day 2 (D2) and Day 8 (D8) post-ovulation in mares susceptible to persistent breeding-induced endometritis. Estrous cycles (*n* = 34) from twelve mares were assigned in a crossover design to receive intrauterine uterine infusions with Lactated Ringer’s Solution (Control, *n* = 12), platelet-rich (PRP, *n* = 12) or -poor plasma (PPP, *n* = 10).

	D2			D8		
Mare ID	Control	PRP	PPP	Control	PRP	PPP
1	*-*	*-*	*-*	*-*	*-*	*-*
2	*-*	*-*	*Enterococcus spp.*	*-*	*-*	*-*
3	*Streptococcus β-hemolytic*	*-*	*-*	*Streptococcus β-hemolytic*	*-*	*-*
4	*-*	*-*	*-*	*-*	*-*	*Klebsiella pneumoniae*
5	*Klebsiella pneumoniae*	*-*	*-*	*Klebsiella pneumoniae*	*-*	*-*
6	*-*	*-*	*-*	*-*	*-*	*-*
7	*Streptococcus β-hemolytic*	*-*	*Streptococcus β-hemolytic*	*-*	*-*	*-*
8	*-*	*-*	*-*	*Escherichia coli*	*-*	*-*
9	*-*	*-*	*Streptococcus β-hemolytic*	*Escherichia coli*	*-*	*Streptococcus β-hemolytic*
10	*-*	*-*	NP	*-*	*-*	NP
11	*-*	*-*	NP	*Escherichia coli*	*-*	NP
12	*-*	*-*	*-*	*-*	*-*	*-*

NP, not performed.

**Table 3 antibiotics-10-00490-t003:** Embryos obtained from mares susceptible to persistent breeding-induced endometritis. Estrous cycles (*n* = 34) from twelve mares were assigned in a crossover design to receive intrauterine uterine infusions with Lactated Ringer’s Solution (Control, *n* = 12), platelet-rich (PRP, *n* = 12) or -poor plasma (PPP, *n* = 10). Embryo flushing was performed eight days after ovulation, with 4 L of LRS. All embryos recovered were measured and graded for development (e.g., blastocyst, or expanded blastocyst) and quality [28].

	Control			PRP			PPP		
Mare ID	Embryo STAGE	Quality Grade	Diameter (µm)	Embryo Stage	Quality	Diameter (µm)	Embryo Stage	Quality Grade	Diameter (µm)
1	-	-		Blastocyst	1	340	Blastocyst	1	260
							Expanded blastocyst	1	1100
2	-	-		-	-		-	-	
3	-	-		Expanded blastocyst	1	1230	Early blastocyst	1	175
4	Expanded blastocyst	1	540	Expanded blastocyst	1	860	Expanded blastocyst	1	840
5	-	-		-	-		-	-	
6	Early blastocyst	2	195	Expanded blastocyst	1	1360	-	-	
7	Blastocyst	1	960	Early blastocyst	2	190	Expanded blastocyst	1	940
8	-	-		Blastocyst	1	340	Expanded blastocyst	1	650
9	-	-		Blastocyst	1	300	-	-	
10	-	-		Blastocyst	1	360	NP	NP	
11	-	-		Early blastocyst	1	165	NP	NP	
12	Blastocyst	1	360	Blastocyst	1	280	Blastocyst	1	320
				Blastocyst	1	360			

**Grade 1** embryo with a spherical shape, uniform size of blastomeres, color, and texture, with no visible abnormalities. **Grade 2** embryo with slight irregularities in shape, size of blastomeres, color or texture, and can present some extruded blastomeres. **Grade 3** embryo can have a large percentage of extruded blastomeres, partial collapse of blastocele, or moderate shrinkage of trophoblast from zona pellucida. **Grade 4** degenerated embryo with variable advanced stages irregularities. [28]. NP, not performed.

## Data Availability

The original contributions presented in the study are included in the article/Appendix A, further inquiries can be directed to the corresponding author/s.

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
