# Peer review of "Intrauterine Blood Plasma Platelet-Therapy Mitigates Persistent Breeding-Induced Endometritis, Reduces Uterine Infections, and Improves Embryo Recovery in Mares"

_antibiotics, 2021, doi:10.3390/antibiotics10050490_

Round 1

Reviewer 1 Report

The topic of this research is interesting and experiment was correctly designed. Many results were obtained, they are valuable and well documented in attached figures and photos. Treatment/prevention of PBIE using PRP is a promising option for reducing the use of antibiotics, and more and more practitioners are interested in this new method.

Suggestions:

Abstract. Please explain abbreviations: PBIE (line 21) and LRS (line 23).

Discussion: 

1.In my opinion, the results of cytokine concentrations in the uterine fluid are poorly discussed - it should be improved.

2.In each group of mares,  the uterus was frequently flushed with 2L of LRS, including flushing six hours post-breeding and again one day later. Uterine flushing using saline solution or LRS is frequently used as a treatment option for PBIE. In my opinion the possible impact of large volume flushing on obtained results should also be discussed.

Author Response

Response to reviewer 1:

The topic of this research is interesting and experiment was correctly designed. Many results were obtained, they are valuable and well documented in attached figures and photos. Treatment/prevention of PBIE using PRP is a promising option for reducing the use of antibiotics, and more and more practitioners are interested in this new method.

Reply: The authors are grateful for the suggestions made by the reviewer. The authors tried to clarify all points.

Query#1. Abstract. Please explain abbreviations: PBIE (line 21) and LRS (line 23).

Reply: Edited as suggested.

Query# 2. 1.In my opinion, the results of cytokine concentrations in the uterine fluid are poorly discussed - it should be improved.

Reply: These results have been discussed further as suggested.  

Query #3. 2.In each group of mares, the uterus was frequently flushed with 2L of LRS, including flushing six hours post-breeding and again one day later. Uterine flushing using saline solution or LRS is frequently used as a treatment option for PBIE. In my opinion the possible impact of large volume flushing on obtained results should also be discussed.

Reply: This point was further discussed as suggested.

Reviewer 2 Report

The study is well designed and involves appropriate controls, randomization, and blinding where it is required. The objectives and hypotheses are clearly stated and the methods are described in great detail. The statistical analyses conducted are appropriate based on the nature of the data. The results are well presented and discussed. Overall, the manuscript is nicely written with only occasional typographical and grammatical errors. I recommend acceptance of the manuscript after a minor revision.

-Line 21: I suggest "Persistent breeding-induced endometritis (PBIE)" instead of "PBIE" here as this is the first instance you are mentioning the condition.

-"plasma-rich (PRP) or -poor (PPP) in platelets": I don't think this is conveying what the authors intend to convey. I suggest rephrasing it as "platelet-rich (PRP) or -poor (PPP) plasma". Alternatively, it could be rephrased as "plasma rich (PRP) or poor (PPP) in platelets. 

-Line 23: I suggest "-48h and -24h" to be consistent with the figures.

-Line 44-45: I suggest using "Persistent breeding-induced endometritis (PBIE)" here and using the acronym afterwards.

-Line 47-48: "PBIE" instead of "persistent breeding-induced endometritis (PBIE)"

-Line 46: I suggest using "spermatozoa" instead of sperm.

-Line 61: "embryo donors' mares"...replace with "donor mares"

-Line 63: Use "premises"

-Line 68: Use "these have"

-Line 95: "with" instead of "into"

-Line 237: I think the authors meant to write "treatment was considered as an independent variable". Fertility rate is the dependent variable here.

-Line 261-265: This statement must be rephrased as it includes multiple double negatives. I suggest "Intrauterine treatment with plasma mitigated PBIE in susceptible mares as evidenced by the reduction of intraluminal and endometrial PMNs, uterine inflammatory cytokines, intrauterine fluid accumulation and the number of positive bacterial cultures compared to control-assigned cycles.

-Line 270: "has been shown"

-Line 329" "an early study indicated that"

Author Response

Response to reviewer 2

The study is well designed and involves appropriate controls, randomization, and blinding where it is required. The objectives and hypotheses are clearly stated and the methods are described in great detail. The statistical analyses conducted are appropriate based on the nature of the data. The results are well presented and discussed. Overall, the manuscript is nicely written with only occasional typographical and grammatical errors. I recommend acceptance of the manuscript after a minor revision.

Reply: The authors appreciated the reviewer’s comments and suggestions. The typographical errors were addressed as suggested.